# Vedolizumab Serum Trough Concentrations and Response to Dose Escalation in Inflammatory Bowel Disease

**DOI:** 10.3390/jcm9103142

**Published:** 2020-09-28

**Authors:** Byron P. Vaughn, Andres J. Yarur, Elliot Graziano, James P. Campbell, Abhik Bhattacharya, Jennifer Y. Lee, Katherine Gheysens, Konstantinos Papamichael, Mark T. Osterman, Adam S. Cheifetz, Raymond K. Cross

**Affiliations:** 1Division of Gastroenterology, Hepatology and Nutrition, Inflammatory Bowel Disease Program, University of Minnesota, Minneapolis, MN 55455, USA; graziano@umn.edu (E.G.); camp0795@umn.edu (J.P.C.); 2Division of Gastroenterology and Hepatology, Medical College of Wisconsin, Milwaukee, WI 53226, USA; ayarur@mcw.edu; 3Division of Gastroenterology, University of Pennsylvania Perelman School of Medicine, Philadelphia, PA 19104, USA; Abhik.Bhattacharya@pennmedicine.upenn.edu (A.B.); Mark.Osterman@pennmedicine.upenn.edu (M.T.O.); 4Division of Gastroenterology, Beth Israel Deaconess Medical Center, Boston, MA 02215, USA; jlee61@bidmc.harvard.edu (J.Y.L.); kpapamic@bidmc.harvard.edu (K.P.); acheifet@bidmc.harvard.edu (A.S.C.); 5Department of Medicine, University of Maryland School of Medicine, Baltimore, MD 21201, USA; KGheysens@som.umaryland.edu; 6Division of Gastroenterology & Hepatology, University of Maryland School of Medicine, Baltimore, MD 21201, USA; RCROSS@som.umaryland.edu

**Keywords:** Crohn’s disease, ulcerative colitis, therapeutic drug monitoring

## Abstract

Serum vedolizumab concentrations are associated with clinical response although, it is unknown if vedolizumab concentrations predict response to dose escalation. The aim of this study was to identify if vedolizumab trough concentrations predicted the response to vedolizumab dose escalation. We assessed a retrospective cohort of patients on maintenance vedolizumab dosing at five tertiary care centers with vedolizumab trough concentrations. Multivariate logistic regression was used to control for potential confounders of association of vedolizumab concentration and clinical status. Those who underwent a dose escalation were further examined to assess if vedolizumab trough concentration predicted the subsequent response. One hundred ninety-two patients were included. On multivariate analysis, vedolizumab trough concentration (*p* = 0.03) and the use of immunomodulator (*p* = 0.006) were associated with clinical remission. Receiver operator curve analysis identified a cut off of 7.4 μg/mL for clinical remission. Of the fifty-eight patients with dose escalated, 74% of those with a vedolizumab concentration <7.4 μg/mL responded versus 52% of those with a vedolizumab trough concentration ≥7.4 μg/mL (*p* = 0.08). After adjustment for relevant confounders, the odds ratio for response with vedolizumab concentration <7.4 μg/mL was 3.7 (95% CI, 1.1–13; *p* = 0.04). Vedolizumab trough concentration are associated with clinical status and can identify individuals likely to respond to dose escalation. However, a substantial portion of patients above the identified cut off still had a positive response. Vedolizumab trough concentration is a potentially helpful factor in determining the need for dose escalation in patients losing response.

## 1. Introduction

Vedolizumab (VDZ), a monoclonal antibody to α_4_β_7_, is effective for induction and maintenance of remission of both Crohn’s disease (CD) and ulcerative colitis (UC) [1,2]. Despite an initial response, incidence rates for loss of response range from 39.8/100 person years for UC to 47.9/100 person years for CD. For anti-tumor necrosis factors (TNF) medications, therapeutic drug monitoring (TDM) has become an effective strategy to manage loss of response. TDM can guide dose escalation when the serum concentration is low, changing to an alternative anti-TNF in the setting of anti-drug antibodies, or changing to a different class of therapy if the concentration is considered adequate [3,4,5,6,7]. The practice of reactive TDM for anti-TNF medications is well established and overall recommended based on a recent guideline statement [8,9]. However, it remains unclear if the principles of TDM can be applied to newer biologics with different mechanisms of action.

Similar to anti-TNFs, VDZ exposure concentrations are associated with clinical response [10]. Infliximab is lost in the stool as part of a protein losing colopathy [11], and as a monoclonal antibody, VDZ may have similar losses, resulting in low trough concentrations in the setting of active inflammation. However, unlike anti-TNF medications, it is unknown if low trough concentration can predict response to dose escalation. The immunogenicity of VDZ is also substantially lower than the anti-TNFs. Up to 30% of loss of response to infliximab in the first year is related to anti-drug antibodies [12]. In contrast, only about 1% of individuals developed persistently positive anti-drug antibodies to VDZ in the pivotal GEMINI clinical trials [1,2]. It is therefore unclear if reactive TDM is a useful strategy in the setting of loss of response to VDZ. Despite this, in patients who experience loss of response to VDZ, 50% will experience a clinical response following a dose escalation, suggesting there is be a relationship to drug exposure [13]. The aim of this study was to determine if VDZ trough concentrations could predict response to dose escalation in the setting of loss of response to VDZ. We hypothesized that individuals with low serum VDZ trough concentrations would be most likely to respond to dose escalation.

## 2. Experimental Section

### 2.1. Patient Population

We performed a multicenter retrospective cohort study in five academic, tertiary care centers (IRB approved at each site or centrally ceded per institutional protocol). We included patients with IBD on maintenance VDZ therapy who had at least one VDZ concentration measured as standard of care. Patients were selected by each site running a report of VDZ serum concentrations ordered by any IBD provider at that site between 2015 and 2019. Charts were manually reviewed for demographics, IBD disease history, medication history, and indication for VDZ concentration testing. The dose escalation population was identified as those who had evidence of loss of response clinically as interpreted by the treating provider on standard dose maintenance VDZ (300 mg every 8 weeks), with an interpretable trough VDZ concentration (i.e., within seven days prior to infusion).

### 2.2. Exposures of Interest

VDZ concentration measurements were made per the local institutional lab agreements. A total of four labs were used to measure VDZ concentrations: Prometheus^®^ Anser^®^ VDZ (liquid-phase mobility shift assay), OptimAbs (enzyme-linked immunosorbent assay), and ARUP and LabCorp (Electrochemiluminescence Immunoassay). Clinical remission at the time of TDM was determined through chart abstraction and interpretation of the treating providers assessment of remission. When available, serologic, endoscopic, radiographic, and fecal surrogate markers were evaluated to supplement the clinical impression. Loss of response/partial response to VDZ was defined as symptoms attributable to underlying IBD in the maintenance phase of VDZ. Loss of response/partial response was further subcategorized as clinical symptoms attributable to underlying IBD or subclinical disease, with subclinical disease including serologic inflammation (elevated C-reactive protein (CRP)), fecal biomarkers, imaging, or endoscopy as available. Reactive TDM was defined as a serum concentration drawn in the setting of loss of response/partial response as defined above. TDM was still considered reactive if the patient was in clinical remission but had evidence of subclinical disease. Dosing changes and response to dosing changes were evaluated through a review of the medical record.

### 2.3. Outcome of Interest

The main outcome of interest was clinical response following dose escalation for patients on maintenance eight-week VDZ dosing. We first established a cohort of patients with VDZ trough concentrations to assess the association of VDZ trough concentration to clinical status. We then examined the subset who had an interpretable VDZ trough concentration and underwent dose escalation on standard maintenance dosing. Response to dose escalation was defined as provider assessed improvement in symptoms, or improvement of the subclinical disease parameter that was abnormal if the patient had no symptoms. For example, a patient who underwent a dose escalation for ongoing endoscopic disease without symptoms was considered a responder if there was improvement of endoscopic disease on follow-up. Improvement was assessed by the treating provider as documented in the medical record and review of primary data. Clinical response was assessed after at least one infusion at the escalated dose. Follow-up assessments were at the discretion of the treating provider. The primary aim of the study was to determine if VDZ trough concentration on standard maintenance dosing predicted clinical response following dose escalation.

### 2.4. Statistical Analysis

Continuous data was summarized as means with SD or median and IQR as appropriate based on normality of data and analyzed with student *t*-test, Wilcoxon test or logistic regression, while categorical data was analyzed by Chi-squared or Fisher’s exact test as appropriate. Logistic regression was used to (1) explore the relationship between VDZ concentration and clinical remission in the total population and (2) determine a VDZ concentration threshold predicted clinical response following dose escalation. For regression analysis VDZ was included as a continuous variable rather than a categorization as it was assumed that the risk of remission would not be homogenous with a various range of VDZ. Both logistic regression models were constructed using an explanatory approach to understand the relationship between VDZ concentration and clinical response following dose escalation. As such, confounding was considered significant if the estimate of the odds ratio for VDZ concentration on clinical response changed by >10%. The final regression model included potential confounders or any variable with a *p* < 0.1 on univariate analysis. Relevant first order interactions were assessed. A *p* value < 0.05 was considered statistically significant.

To determine if VDZ concentration predicted clinical remission at the time of TDM, multiple logistic regression was performed controlling for age, sex, disease type, VDZ dosing interval, prior biologic exposure status, immunomodulator use, VDZ treatment duration, and disease duration. For the purpose of regression analysis patients with indeterminate colitis were classified as UC. Steroid use was not incorporated into this model due to confounding by indication. Assay type (Prometheus, OptimAbs, other) was included in the final model to control for inter-assay variation. A number of sensitivity analyses were performed including the effect of VDZ concentration on endoscopic remission and the two most common assay types individually (Prometheus and OptimAbs). Additionally, as VDZ is a protein and the serum concentration may reflect underlying protein metabolism, a sensitivity analysis was performed controlling for albumin on the effect of VDZ concentration and clinical remission. Receiver operating curves (ROC) were used to determine cut off values of VDZ to predict clinical remission (based on Youden index). VDZ was then incorporated as a binary low/high variable to assess if low VDZ trough concentration predicted clinical response to dose escalation. Multivariate logistic regression was performed to assess for potential confounders as well as independent predictors of clinical response including age, sex, disease type, prior biologic exposure status, immunomodulator use, steroid use (≥20 mg of prednisone or equivalent), VDZ treatment duration, and disease duration. Statistics were performed in JMP^®^ Pro v14.0.0, multiple regression was performed in R v3.6.0.

## 3. Results

Two hundred and thirty-nine VDZ serum concentrations were initially identified. Forty-five were non trough concentrations and two had missing VDZ data resulting in 192 unique VDZ trough concentrations for analysis. Demographics of the cohort are presented in Table 1. No antibodies to VDZ were identified. Thirty-two percent (61) individuals were in clinical remission at the time of TDM.

### 3.1. Vedolizumab Correlation with Clinical Remission

Overall, VDZ trough concentration was significantly associated with VDZ dosing interval (Figure 1). Those on q4 week dosing interval had a median trough concentration of 30.5 μg/mL, IQR 20.2, 39.4, which was higher than those on q5–7 week dosing (median: 13.9 μg/mL, IQR 7.4, 25.6, *p* = 0.05) and q8 week dosing (median: 8.9 μg/mL, IQR 5, 13.6, *p* < 0.0001). There was no variation of VDZ concentration by immunomodulator status within a given dosing interval. Multiple logistic regression was used to assess for an independent effect of VDZ trough concentration controlling for key clinical factors and other independent predictors of remission. Overall, VDZ trough concentration was significantly associated with clinical remission at the time of TDM testing (OR: 1.03, 95% CI: 1.01, 1.05; *p* = 0.01). Sex, IBD type, and VDZ dosing interval were identified as confounders of the relationship on VDZ trough concentration and clinical remission. In addition to VDZ concentration, IBD type and immunomodulator use were independent predictors of clinical remission. In multivariate analysis, controlling for the type of assay used, VDZ trough concentration and immunomodulator use were associated with clinical remission (Table 2).

When limiting analysis to individuals with endoscopic data available (*n* = 113), VDZ concentration was marginally associated with endoscopic remission (OR: 1.03, 95% CI: 0.99, 1.06; *p* = 0.07). When examining assay subsets by assay type, VDZ was associated with clinical remission (OR: 1.05, 95% CI: 1.01, 1.1; *p* = 0.03) among those with the Prometheus assay, and marginally associated among those with OptimAbs assay (1.03, 95% CI: 0.99, 1.07; *p* = 0.05). Among the entire cohort, when controlling for albumin concentration, VDZ concentration was no longer a significant predictor of clinical remission (OR: 1.01, 95% CI: 0.99, 1.04; *p* = 0.3), while albumin did independently predict clinical remission (OR: 2.9, 95% CI: 1.3, 6.2; *p* = 0.008). Among those with an eight-week trough concentration, ROC analysis identified an optimal cut point of 7.4 μg/mL for predicting clinical remission (AUROC 0.57) and a cut off of 8 μg/mL for endoscopic remission (AUROC: 0.62).

### 3.2. Vedolizumab Trough Concentration and Response to Dose Escalation

Of the 192 VDZ concentrations, 65 underwent a dose escalation on q8 week maintenance dosing and had sufficient clinical follow-up data to determine response. Seven patients who underwent dose escalation were proactive dose adjustments and thus no clinical response was available, leaving 58 reactive dose escalations for evaluation. Of the 58 reactive dose escalations, 83% (48) were due to loss of response/partial, while 10 were due to subclinical disease. The resulting dose of VDZ was 300 mg q4 week intervals for 81% of the cohort, while the remainder were escalated to 300mg every 5–7 weeks. The median VDZ concentration among those who underwent dose escalation was 7.8 (IQR: 3.9, 11.5). The overall response to dose escalation was 62%. The median time to follow-up to assess clinical response was 15 weeks (IQR 9.5, 23.5 weeks). This was similar for those who had a clinical response (median 14 weeks, IQR: 10.5, 22 weeks) and for those who did not (median 15.5 weeks, IQR: 9.5, 23.5 weeks). Among responders to dose escalation, the median VDZ concentration was 7.05 μg/mL (IQR: 3.3, 10.6 μg/mL) while the median VDZ trough concentration among non-responders was 8.2 μg/mL (IQR: 4.3, 12.4 μg/mL). This pattern was similar for CD and UC responders (7.3 μg/mL (IQR: 2.3, 10 μg/mL) and 6 μg/mL (IQR: 3.5, 11.4 μg/mL) respectively), and non-responders (8.2 μg/mL (IQR: 4.1, 15 μg/mL) and 8.2, (IQR: 4.1, 11.2 μg/mL) for CD and UC respectively).

Using the identified optimal cut off of 7.4 μg/mL, 74% of patients with an eight-week VDZ trough concentration below the cutoff achieved response following dose escalation, versus 52% whose trough was ≥7.4 μg/mL (Figure 2, *p* = 0.08). VDZ response by quartile is presented in Figure 3. No significant confounding was noted with sex, prior biologic use, VDZ duration, disease duration or immunomodulator use at the time of VDZ trough measurement, while age changed the estimate of the OR by more than 10%. Additionally, steroid use at the time of TDM was associated with clinical response following dose escalation. Notably, albumin at the time of TDM was not associated with clinical response following VDZ dose escalation. Dose escalation to q4 week intervals (versus q5–7) did not predict response. There was no confounding or effect modification between assay type on VDZ concentration and response. In a multivariable logistic regression model controlling for age and steroid use at the time of TDM, VDZ concentration was a significant independent predictor of response to dose escalation (*p* = 0.04) along with steroid use at time of TDM (Table 3). The OR for VDZ concentration < 7.4 μg/mL responding to dose escalation was 3.7 (95% CI 1.1, 13) compared to those with VDZ level ≥ 7.4 μg/mL. Regression results were unchanged when excluding those with only subclinical disease and controlling for assay type (data not shown).

## 4. Discussion

We performed a large, multicentered, retrospective cohort of IBD patients on maintenance VDZ with the primary aim of determining if VDZ trough concentration on standard maintenance dosing predicted response to dose escalation. We first examined all VDZ trough concentrations to explore the relationship with clinical remission and identify an optimal VDZ cut off in our cohort. We identified both trough VDZ concentration and immunomodulator use to be associated with clinical remission on multivariate analysis. Among those on standard VDZ maintenance dosing who underwent a dose escalation, 62% experienced a response following dose escalation, which is similar to a prior meta-analysis of dose response to VDZ [13]. An eight-week VDZ trough concentration of < 7.4 μg/mL predicted a positive response to dose escalation, although 52% of those above the cut off still responded. Low trough VDZ (<7.4 μg/mL) remained a significant predictor of response to dose escalation after controlling for confounding variables. These findings suggest that VDZ trough concentration is correlated to disease activity, but low trough concentrations alone do not account for the effect of dose escalation.

With anti-TNF agents, low serum trough concentration accurately identifies those who would benefit from dose escalation [3]. Anti-TNF medications work through systemic cytokine suppression. This mechanism may lend itself towards a more traditional dose-response relationship mediated by a serum concentration. VDZ, on the other hand, selectively inhibits α_4_β_7_ integrins expressed by T lymphocytes destined for the gastrointestinal tract, preventing translocation into the intestine [14]. In the GEMINI clinical trials, all doses of VDZ achieved α_4_β_7_ receptor saturation of >95% in the study population [1,2]. If receptor saturation alone were the complete mechanism of action for VDZ, then it is unlikely that dose escalation would result in a clinical benefit. However, recent data suggests that the actual mechanism of action of VDZ is less reliant on inhibition of T cell trafficking and more related to changes in the mucosal innate immune system [15]. VDZ may therefore act through both dose dependent and dose independent mechanisms. Therefore, those with low drug concentrations may benefit from a dose escalation through mechanisms other than α_4_β_7_ receptor saturation. Alternatively, VDZ concentrations may be a biomarker of inflammatory status with lower drug concentrations associated with more inflammation related to increased protein clearance. This is suggested in our data by the lack of association of VDZ level with clinical remission after adjustment for serum albumin. However, albumin did not predict response to VDZ dose escalation, suggesting that while both may be biomarkers of inflammation, VDZ concentration has a mechanistic role as well.

While VDZ trough concentration could predict those likely to respond to dose escalation, many patients above our calculated VDZ trough cut off still had a response. One possible explanation for this finding is that patients in our dose escalation cohort had trough concentrations below a therapeutic cut off. This effect has been noted with anti-TNF agents. For patients on infliximab or adalimumab, while maintenance trough levels of 3–5 μg/mL are associated with clinical remission higher concentrations (10–15 μg/mL) are associated with more rigorous and objective outcome measures, including endoscopic and histological remission, in both UC and CD [16]. Similarly, with VDZ, a post hoc analysis of the pivotal GEMINI 1 randomized controlled trial in UC patients found that VDZ maintenance trough levels of 24–28 μg/mL were associated with higher rates of endoscopic mucosal healing, with the highest quartile having serum concentrations > 42 μg/mL [17]. As the median VDZ concentration among those who underwent dose escalation in our cohort was 7.8 μg/mL, we were unable to adequately assess whether dose escalation would be beneficial using higher thresholds. Ungar et al. presented preliminary data on VDZ trough concentrations in maintenance phase finding that pre-dose intensification concentration was not predictive of response [18]. Notably this study had substantially higher median maintenance phase concentrations for both responders and non-responders. However, in sum, it appears other factors than trough concentration may be responsible for success of dose escalation.

It is also notable that, in our cohort, we did not identify any anti-drug antibodies. An important part of a reactive TDM strategy for anti-TNF medications is the identification of anti-drug antibodies, which guide the decision for dose escalation, change within class, or class change. As VDZ has substantially lower immunogenicity, empiric VDZ dose escalations on standard maintenance dosing for loss of response may be reasonable regardless of trough concentration. VDZ concentrations may be more useful in patients already on non-standard VDZ dosing who have continued disease activity to determine if further dose escalation or changing drug class would be most effective.

Our cohort additionally identified immunomodulator use to be associated with clinical remission independent of VDZ concentration and dosing interval, although response to VDZ dose escalation was not impacted by immunomodulator use. Immunomodulator use can be beneficial for the prevention of anti-drug antibodies with biologics but can also treat the underlying disease. Our data suggest that immunomodulator use is beneficial in achieving remission through immune suppression rather than its effect on VDZ. On the other hand, steroid use at the time of TDM was negatively associated with a response to dose escalation. Patients using high dose steroids at the time of TDM may have more refractory disease, and thus be less likely to experience a clinical response. However, regardless of steroid use, low VDZ concentrations remained predictive of response to dose escalation.

Our study is limited by the inherent nature of a retrospective study. Clinical status (remission and response) were abstracted from clinical documentation and not collected prospectively using validated scoring systems. While supplemental information was used to validate the clinical information as available (e.g., CRP, fecal calprotectin, endoscopy), there was insufficient pre-post dose escalation data available to examine the effect on other inflammatory parameters. Our study also may have been underpowered to identify a smaller effect of VDZ trough concentration on response following dose escalation. This is due in part to the non-standardized dose escalation as 19% of patients were escalated to less than q4 week intervals. However, the 22% improvement in rate of clinical response due to dose escalation for patients under the cut-off VDZ level compared to those above this level is clinical meaningful and may have been biased to the null by insufficient dose escalations. Another limitation of this study is the lack of a control group of patients that underwent dose escalation irrespective of VDZ trough concentration. Thus, there may still be a valuable cut off that is higher than what was calculated in our dose escalation group, as discussed above. However, validation in a prospective cohort with a standardized protocol for dose escalation is needed. Finally, our study only evaluated VDZ concentrations in the maintenance phase of therapy. Proactive TDM beginning during the induction phase of therapy may be valuable to guide dosing, and there is a current clinical trial (ENTERPRET, NCT03029143) to answer this question that is nearly complete at the time of this manuscript submission.

## 5. Conclusions

In conclusion, VDZ serum trough concentration is associated with clinical remission. In our cohort, a VDZ eight-week trough concentration of 7.4 μg/mL predicted a positive response to dose escalation. While VDZ trough concentration was associated with response to dose escalation, approximately half of individuals above our cut off still had a positive response. VDZ maintenance trough concentrations may help predict patients who will respond to dose escalation, but other factors also likely contribute.

## Figures and Tables

**Figure 1 jcm-09-03142-f001:**
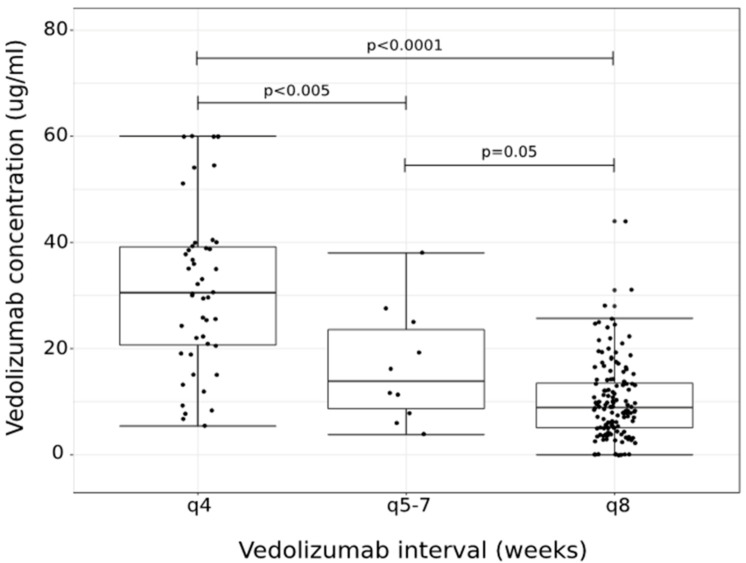
Vedolizumab (VDZ) trough concentration stratified by dosing interval. Shortened VDZ dosing intervals were significantly associated with increased serum VDZ concentrations.

**Figure 2 jcm-09-03142-f002:**
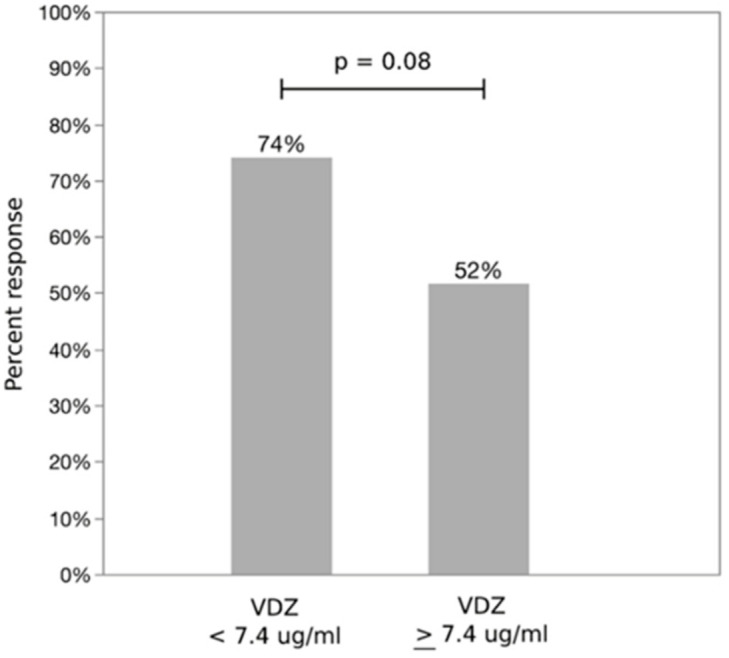
Proportion of responders following Vedolizumab (VDZ) dose escalation using cut off identified by ROC analysis of 7.4 ug/mL. VDZ trough was marginally associated with response to dose escalation.

**Figure 3 jcm-09-03142-f003:**
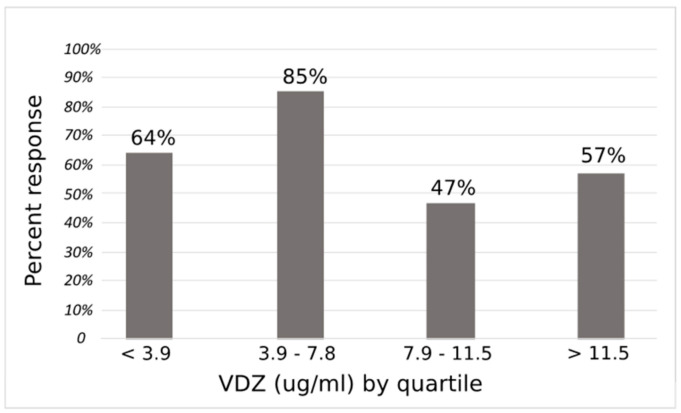
Proportion of responders following Vedolizumab (VDZ) dose scalation by VDZ quartile.

**Table 1 jcm-09-03142-t001:** Baseline characteristics.

	N (%) ^1^
Female	103 (54)
Median Body mass index (IQR)	26.1 (23, 30.7)
Median age at VDZ testing (IQR)	41 (29, 60)
Median disease duration (IQR)	10 (5, 19)
IBD type	
Crohn’s disease	87 (45)
Ulcerative colitis	94 (49)
IBD-U	11 (6)
Never smoker	123 (64)
Bio-naïve prior to VDZ	50 (26)
Prior anti-TNF use	141 (77)
Prior anti-IL12/23	16 (9)
VDZ dose interval at TDM	
Q8 weeks	136 (71)
Q5–7 weeks	10 (5)
Q4 weeks	47 (24)
IMM use at TDM	54 (28)
Thiopurine	31 (57)
Methotrexate	20 (37)
Other	3 (6)
Steroid use at TDM ^2^	28 (14)
Clinical remission at TDM	61 (32)
Endoscopic remission at TDM ^3^	32 (28)
TDM Assay	
Prometheus	87 (45)
OptimAbs	81 (42)
Other ^4^	24 (13)

VDZ: Vedolizumab; IQR: Interquartile range; IBD: Inflammatory Bowel Disease; TNF: Tumor necrosis factor; IL: interleukin; TDM: Therapeutic drug monitoring, ^1^ Percent unless otherwise specified, ^2^ Prednisone ≥ 20 mg (or equivalent), ^3^ Data available for 113/192, ^4^ Other includes ARUP and LabCorp.

**Table 2 jcm-09-03142-t002:** Logistic regression analysis of clinical remission at the time of TDM.

	Univariate		Multivariate ^1^	
	OR (95% CI)	*p* Value	OR (95% CI)	*p* Value
Age	1.00 (0.98, 1.02)	0.9		
Male sex ^2^	1.4 (0.78, 2.6)	0.2	1.5 (0.74, 3.1)	0.3
Ulcerative colitis/IBD-U ^2^	1.75 (0.93, 3.26)	0.08	1.8 (0.87, 3.6)	0.1
VDZ interval [ref: q4 week dosing] ^2^				
q5–7 week dosing	1.56 (0.39, 6.14)	0.5	2.3 (0.45, 11.9)	0.3
q8 week dosing	0.60 (0.30, 1.21)	0.2	1.3 (0.44, 3.9)	0.6
Prior biologic use	0.69 (0.33, 1.41)	0.3		
IMM use at TDM	2.7 (1.4, 5.3)	0.003	2.8 (1.3, 5.7)	0.006
VDZ treatment duration	1.01, (0.99, 1.03)	0.2		
Disease duration [ref: <4 years]				
4–7 years	2.04 (0.67, 6.3)	0.2		
>8 years	1.6 (0.62, 4.4)	0.2		
VDZ concentration	1.02 (1.01, 1.05)	0.01	1.03 (1.004, 1.08)	0.03

IMM: Immunomodulator; TDM: Therapeutic drug monitoring; VDZ: Vedolizumab. IBD-U: Inflammatory bowel disease–unspecified; IMM: Immunomodulator; TDM: Therapeutic drug monitoring, VDZ: Vedolizumab, ^1^ Final model controlled for TDM assay type. ^2^ Confounder between VDZ concentration and clinical remission at the time of TDM. Confounding defined as a >10% change in the estimate of the odds ratio of VDZ on clinical remission.

**Table 3 jcm-09-03142-t003:** Multivariate regression to predict clinical response to reactive dose escalation.

	Univariate OR	*p* Value	Multivariate OR ^1^	*p* Value
Age ^2^	1 (0.98, 1.04)	0.6	1 (0.98, 1.06)	0.3
Male sex	1.2 (0.41, 3.5)	0.7		
Ulcerative colitis ^3^	1.3 (0.40, 3.8)	0.7		
Prior biologic use	1.1 (0.28, 4.2)	0.9		
IMM use at TDM	1.8 (0.42, 7.7)	0.4		
Steroid use at TDM ^4^	0.22 (0.056, 0.85)	0.03	0.16 (0.03, 0.71)	0.01
VDZ treatment duration	0.98 (0.94, 1.03)	0.5		
Disease duration [ref < 4 years]				
4–7 years	1.8 (0.21, 15.4)	0.9		
≥8 years	3.2 (0.48, 21.7)	0.2		
VDZ concentration < 7.4 μg/mL	2.7 (0.88, 8.2)	0.08	3.7 (1.1, 13)	0.04

IMM: Immunomodulator; TDM: Therapeutic drug monitoring; VDZ: Vedolizumab, ^1^ Final model controlled for TDM assay type. ^2^ Confounder between VDZ concentration and response to dose escalation. ^3^ Excluded single case of IBDU for selected analysis, ^4^ Prednisone ≥ 20 mg (or equivalent).

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
