# Peer review of "Vedolizumab Serum Trough Concentrations and Response to Dose Escalation in Inflammatory Bowel Disease"

_jcm, 2020, doi:10.3390/jcm9103142_

Round 1
Reviewer 1 Report
I suggest the authors need to clarify what OR stands for in the abstract; otherwise I support the presentation as it is a useful adjunct to clinical care.
Author Response
We appreciate the reviewers comments. A point by point response is below:
|
I suggest the authors need to clarify what OR stands for in the abstract; otherwise I support the presentation as it is a useful adjunct to clinical care.
|
We have updated the abstract to odds ratio as opposed to OR. |
Reviewer 2 Report
This paper is a multicentre retrospective cohort study examining therapeutic drug monitoring in adult patients treated with vedolizumab for inflammatory bowel disease.
The authors found that there was a weak, but statistically significant, association between trough VDZ levels and clinical remission. Combination therapy with an immunomodulator was also, more strongly, associated with clinical remission. For 58 patients who underwent reactive VDZ dose escalation, 62% had improvement of symptoms or disease markers after a median time of 15 weeks. Approximately three-quarters of patients with a trough VDZ level below 7.4 µg/mL responded to dose escalation, whereas half of the patients with trough VDZ above that level responded.
This well-written paper adds to a growing literature seeking to place therapeutic drug monitoring in clinical practice – something already achieved with some biologic drugs. Algorithm-based treatment of IBD, incorporating TDM, may offer superior care to patients, and this paper shows that trough levels may be a moderate predictor of treatment response after dose escalation. The study examines real-world use of vedolizumab in a mixed IBD cohort. The analysis and interpretation are appropriate and the discussion is considered. I would like to make a few minor suggestions:
- As the authors indicate in the introduction, durable remission rates differ for ulcerative colitis and Crohn’s disease; although it does not seem from table 3 that type of IBD was a significant variable in determining treatment response, it may be useful to see data or a figure demonstrating this. For example – graphing VDZ levels vs. a) Crohn’s disease response, b) Crohn’s disease non-response, c) UC response, d) UC non-response. Is the cut-off VDZ level different for Crohn’s disease compared to UC?
- The limitations of the study are addressed, but it is important to note that the findings should be validated in a prospective cohort
- In figure 1, the y-axis should start at zero
- “c reactive protein” should be capitalised (“C-reactive protein”)
Author Response
We appreciate the reviewers comments. A point by point response is below:
|
As the authors indicate in the introduction, durable remission rates differ for ulcerative colitis and Crohn’s disease; although it does not seem from table 3 that type of IBD was a significant variable in determining treatment response, it may be useful to see data or a figure demonstrating this. For example – graphing VDZ levels vs. a) Crohn’s disease response, b) Crohn’s disease non-response, c) UC response, d) UC non-response. Is the cut-off VDZ level different for Crohn’s disease compared to UC? |
VDZ trough concentrations for UC and CD were very similar between responders and non-responders. This data was added as requested to the text of the results. |
|
The limitations of the study are addressed, but it is important to note that the findings should be validated in a prospective cohort |
We agree with the reviewer that these findings require validation. This has been added to the limitations section of the discussion. |
|
In figure 1, the y-axis should start at zero |
Due to the number of undetectable VDZ concentrations at q8 week dosing, the Y axis is slightly offset so that the lower box and whisker plot line can be seen. Our preference is to maintain the current figure so the reader can appreciate all three box plots side by side. |
|
“c reactive protein” should be capitalised (“C-reactive protein”) |
We thank the review for the attention to detail and this change has been made. |
Reviewer 3 Report
In this manuscript, the authors analyzed vedolizumab levels and response to dose escalation in IBD. They showed that vedo trought concentration should be potentially an helpful factor in determining the need for dose escalation in patients losing response. This study is important for the clinical practice
major comments :
what is the new dose of vedolizumab after escalation (e4W for all) ? report this point and discuss if dose escalation was not the same of all.
Should be necessary to describe shortly the assays used in this study.
In the introduction , the authors reported that VDZ could be lost in the stool using ref from vande casteele for IFX (11). No clear data with vedo and stool losing. Modify this point
Some limitions due to the retrospective design and definition of outcome of interest but the authors discussed clearly these limitations. hav ethe authors some data about fecal calpro after optimization?
Should be interesting to report clinical remission according to VDZ levels before optimization using quartiles of VDZ.
Report type of IS drugs and dose. Report also BMI
Have the authors some data about through concentrations of VDZ afetr optimization?
discuss poster presentation on ECCO and DDW 2020 by Ungar et al reported conflicting data with no correlationbetween through concentration of VDZ before optimzation and response after optimization. Discuss this point.
Author Response
We appreciate the reviewers critiques and believe these changes have improved the quality of this manuscript. A point by point response is below:
|
what is the new dose of vedolizumab after escalation (e4W for all) ? report this point and discuss if dose escalation was not the same of all. |
The reviewer raises an excellent point. 80% of the dose escalations were to e4W dosing, while the other 20% were a variation of q5-7 weeks. Escalation to q4 weeks did not predict response (v. q5-7 weeks). However, this is clearly a limitation of this retrospective cohort and a standardized protocol of dose escalation in a prospective cohort would better address this. This information was added to the results and the limitations. |
|
Should be necessary to describe shortly the assays used in this study. |
Commercial labs were used per local practice and assay type added to methods. |
|
In the introduction , the authors reported that VDZ could be lost in the stool using ref from vande casteele for IFX (11). No clear data with vedo and stool losing. Modify this point |
The reviewer is correct this citation is for IFX and we are extrapolating this finding to VDZ. This point has been clarified. |
|
Some limitions due to the retrospective design and definition of outcome of interest but the authors discussed clearly these limitations. hav ethe authors some data about fecal calpro after optimization? |
We agree with the reviewer on these limitations. Unfortunately, fecal calprotectin was not widely used over the study period due to insurance coverage issues. While data was requested regarding this, there were too few samples to make any meaningful descriptive data. |
|
Should be interesting to report clinical remission according to VDZ levels before optimization using quartiles of VDZ. |
This has been added as Figure 3. |
|
Report type of IS drugs and dose. Report also BMI |
BMI and thiopurine at the time of TDM has been added to demographics table. Unfortunately data on IMM dose was not collected. |
|
Have the authors some data about through concentrations of VDZ afetr optimization? |
We agree with the reviewer that this would be important information to capture. Understanding the effect on concentration following dose escalation is an important question. Unfortunately, we do not have data after dose escalation as these concentrations were drawn clinically. |
|
discuss poster presentation on ECCO and DDW 2020 by Ungar et al reported conflicting data with no correlation between through concentration of VDZ before optimzation and response after optimization. Discuss this point. |
We appreciate the attention to this citation. While we did identify a pharmacokinetic cut off, we do identify it as weak, and overall the clinical impact is unclear. Indeed, we overall note that many people with higher concentrations still responded. Taken with the preliminary Ungar data (now included in the discussion) it is likely that other factors apart from serum drug concentration account for the effect of dose escalation. |